# Vegetation Loss Measurements for Single Alley Trees in Millimeter-Wave Bands

**DOI:** 10.3390/s24103190

**Published:** 2024-05-17

**Authors:** Krzysztof Cichoń, Maciej Nikiforuk, Adrian Kliks

**Affiliations:** 1Institute of Radiocommunications, Poznan University of Technology, 60-965 Poznań, Poland; krzysztof.cichon@put.poznan.pl; 2Tietoevry, al. Piastów 30, 71-064 Szczecin, Poland; maciej.nikiforuk@tietoevry.com; 3Department of Computer Science, Electrical and Space Engineering, Luleå University of Technology, 971 87 Luleå, Sweden

**Keywords:** foliage attenuation, millimeter-wave propagation, vegetation attenuation, radiative transfer equation

## Abstract

As fixed wireless access (FWA) is still envisioned as a reasonable way to achieve communications links, foliage attenuation becomes an important wireless channel impairment in the millimeter-wave bandwidth. Foliage is modeled in the radiative transfer equation as a medium of random scatterers. However, other phenomena in the wireless channel may also occur. In this work, vegetation attenuation measurements are presented for a single tree alley for 26–32 GHz. The results show that vegetation loss increases significantly after the second tree in the alley. Measurement-based foliage losses are compared with model-based, and new tuning parameters are proposed for models.

## 1. Introduction

Fixed wireless access is a promising technology for fast-speed internet access without the need for costly excavation to connect optical fibers. In general, in FWA, two points with fixed locations are connected in a wireless way, and the major considered frequency band is a millimeter-wave band (mmWave). Furthermore, in mmWave, there is a great amount of bandwidth, which allows high user throughput. Moreover, it is possible that in an FWA link, there may be no line of sight, and the presence of foliage is possible.

There are examples where foliage is an important factor to consider in mmWave links. First, dedicated software for predicting path loss in a foliage-dominant environment is described in [1]. It utilizes the empirical model P.1812 with a physics-based radiative energy transfer. Second, the prediction of UAV link coverage is characterized in simulation-based work [2]. The probability of coverage was shown to be reduced for 28 and 60GHz by up to 13% and 43%, respectively, if the attenuation of foliage and rain is taken into account. Therefore, the attenuation of foliage is an important impairment that must be considered in modern systems. In addition, many measurement campaigns have been carried out in different frequency bandwidths [3,4,5,6,7].

For instance, in [8], measurements were made for LoRa transmission at a frequency of 868MHz in an artificial forest consisting of three thousand elm, ash, and plane trees. The path loss results presented showed a good fit with empirical models. Furthermore, the authors claim that the dispersion of the results increases with the distance between the transmitter and receiver, and for 50m and 200m, it is 12dB and 18dB, respectively.

Other articles have investigated the effects of foliage in millimeter-wave bands. In [9,10], the examination focused on the frequency of 28GHz. Two different measurement scenarios were prepared and investigated, one for the parallel orientation of the measurement link to the tree alley and the other for the skew orientation. After the analysis of the measurements, it was found that two types of components could be distinguished in the receiving signal: a well-defined directive component and extra diffusive components. Furthermore, the authors compared their results with various vegetative attenuation models, including the models in the document ITU-P.833. Based on the measurement points, the appropriate coefficients were tuned and derived. In [10], the seasonal variability of vegetation for the path loss measurements was verified. The measurements were carried out during December when leaves barely covered the trees and in October when the leaves were lush.

The authors of this paper also made some measurements of vegetation loss, described in [11], where two different scenarios were considered. In the first scenario, the attenuation of the London plane tree was verified, with the transmitter located at a height of 5m and the receiver located at a height of 3.6m. This is the typical scenario for cellular networks, where the base station is of greater height compared to the user equipment (UE). On the other hand, in another scenario (referred to as Scenario 3), the transmitter and receiver were placed at the same height of 2.6m; therefore, parallel orientation was considered. Measurements were made in the vicinity of small trees in the park, namely, London plane trees. The average diameter of the corona was 1.4m; however, the branches inside the corona were rather dense. This may be the reason why the attenuation of the foliage in that scenario was claimed to be 10dB higher than for regular trees and nonhorizontal measurements [11]. It was somewhat surprising that the parallel scenario was the worst-case scenario. Thus, in this work, the parallel scenario, as is considered for fixed wireless links, is considered to be the main use case for further investigation.

As new frequency bands are envisioned for future communication systems, including mmWaves, sub-terahertz, and terahertz, a new investigation is needed. In the terahertz domain, there are some interesting works related to speckle patterns [12], spatiotemporal wave synthesis [13], and sampling of the terahertz field [14]. In sub-terahertz, conducted measurements in the urban micro scenario showed the foliage loss to be no greater than 11dB [15]. Similarly, in mmWave, the loss of vegetation is elaborated in [16]. In the series of measurements at 110–170 GHz, various categories of foliage were checked: for example, forest, single tree, hedge, tree trunk, and various species. Based on the measurement data, the authors extended the maximum exponential decay (MED) model by adding a new parameter, namely, the plant area index (PAI). This parameter was used to characterize the density of the foliage. The root mean square error (RMSE) between the measured PL data and the fit of the model was 5.6dB.

Taking into account the above analysis, we see that further measurements are needed. We observed that in the literature, foliage is considered in two ways: (i) as a continuous woodland, according to the ITU-R.P.833 document [17], or (ii) as a single tree. Thus, in our work, we concentrate on the new case, where we carry out measurements of foliage loss in a tree alley. In principle, our measurement points are located between trees in the tree alley at the same distance.

This paper is organized as follows. In the next section, additional attenuation in vegetation is analyzed based on the radiative transfer equation. In Section 2.2, the tree alley perspective is analyzed, and later on, the statistical models are described in Section 2.3. The measurement campaign is described in detail in Section 3. The measurement results are presented in Section 4, followed by discussion in Section 5. The paper is summarized and conclusions are drawn in Section 6.

## 2. Additional Attenuation of Vegetation

### 2.1. Radiative Transfer Equation Perspective

Foliage can be modeled as a medium of random scatterers where the electromagnetic field has two components: coherent and incoherent [12,18,19]. The coherent component is the average field with a known direction of propagation and a defined polarization. It is susceptible to both absorption and scatter, and as a result, the attenuation of this component is high. The incoherent component is the field with zero mean, which is the result of the scattering of the coherent component. Therefore, it has no specific direction and is partially depolarized.

In transport theory [20], the fundamental quantity is the specific intensity $I(\bar{r},\bar{s})$ [21], defined as
(1)$$I\left(\bar{r},\bar{s}\right)=\frac{d^{2}P\left(\bar{r},\bar{s}\right)}{dad\Omega }.$$

It is defined as the power slice $d^{2}P$ per unit area *a* and per unit solid angle Ω (compare with Figure 1) propagating at the point $\bar{r}$ in the direction $\bar{s}$.

Absorption and scattering by the woodland medium reduce the power propagating in the direction $\bar{s}$:(2)$$dI_{1}=-\left(\sigma_{a}+\sigma_{s}\right)I\left(\bar{r},\bar{s}\right)dl,$$
where $dI_{1}$ is the instantaneous power reduction rate, $\sigma_{a}$ is the absorption cross section per volume, and $\sigma_{s}$ is the scatter cross section per volume. *l* is the length such that volume dV=da·dl. A fraction of the power propagating in other directions ${\bar{s}}^{\prime }$ will contribute to the main propagating component in the direction $\bar{s}$. Then, the increase of intensity $I(\bar{r},\bar{s})$ by fraction of the power specific for the medium $p(\bar{s},{\bar{s}}^{\prime })$ is equal to
(3)$$dI_{2}=\frac{1}{4\pi }\sigma_{s}\int \int_{4\pi }p\left(\bar{s},{\bar{s}}^{\prime }\right)I\left(\bar{r},{\bar{s}}^{\prime }\right)\;d\Omega^{\prime }\;dl,$$
where $dI_{2}$ describes the change in the contribution from the so-called medium’s backlighting, $p(\bar{s},{\bar{s}}^{\prime })$ is the *phase function* of the scattering medium that describes the angular distribution of power scattered by the medium, and $d\Omega^{\prime }\;$ is a dummy integration variable. It is the differential element of the solid angle used for the purpose of the integration over all spatial directions ${\bar{s}}^{\prime }$. Since the reciprocity of the channel ensures that the same results will be obtained when the transmitter and receiver are interchanged, therefore $p\left(\bar{s},{\bar{s}}^{\prime }\right)=p\left({\bar{s}}^{\prime },\bar{s}\right)$. Then, the total variation of the intensity *I* over length dl is the following:(4)$$dI=\left[\bar{s}\cdot \mathrm{grad}\left[I\left(\bar{r},\bar{s}\right)\right]\right]dl=dI_{1}+dI_{2},$$
so it is the sum of intensities from Equations (Equation 2) and (Equation 3) where $\mathrm{grad}[I\left(\bar{r},\bar{s}\right)]$ is the gradient of the intensity function at point $\bar{r}$ and direction $\bar{s}$. Hence, when (Equation 2) and (Equation 3) are put in (Equation 4), then:(5)$$\bar{s}\cdot \mathrm{grad}\left[I\left(\bar{r},\bar{s}\right)\right]+\left(\sigma_{a}+\sigma_{s}\right)I\left(\bar{r},\bar{s}\right)=\frac{\sigma_{s}}{4\pi }\int \int_{4\pi }p\left(\bar{s},{\bar{s}}^{\prime }\right)I\left(\bar{r},\bar{s}\right)\;d\Omega^{\prime }$$

Equation (Equation 5) is known as the scalar form of the transport equation.

When modeling the attenuation in the foliage, the simplified model can be taken into account. The woodland is a homogeneous medium of random scatterers, where the scatter elements are large compared to the wavelengths in the millimeter-wave bandwidth. Then, the forward-scattering component p(γ) can be modeled as a Gaussian lobe with a width of $\Delta \gamma_{s}$ (Figure 2):
(6)$$p\left(\gamma \right)=\alpha {\left(\frac{2}{\Delta \gamma_{s}}\right)}^{2}e^{{-\left(\frac{\gamma }{\Delta \gamma_{s}}\right)}^{2}}+\left(1-\alpha \right),$$
where γ represents angle deviation to the main axis γ=0, $\Delta \gamma_{s}$ is the beamwidth, and α is the forward-to-total scattered power ratio. As the transmitter is located sufficiently far from the woodland edge, the intensity entering the medium with zero boundary conditions for the *z* axis can be represented as
(7)$$\begin{array}{c} I\left(0;\theta ;\phi \right)=S_{p}\frac{\delta (\theta -\theta_{p})}{\sin\theta_{p}}\delta \left(\phi -\phi_{p}\right), \end{array}$$
where δ(·) is the delta function and $S_{p}$ is the Poynting vector, while $\theta_{p}$ and $\phi_{p}$ are the angles of the incident intensity and θ, ϕ are the angle counted from the positive Z-direction and the projected angle in planes Z=constant, respectively. Please note the formula is valid for θ∈(0,π/2). Specific intensity consists of two parts, as shown in (Equation 8). The first one, $I_{\mathrm{ri}}$, represents the reduced intensity connected with the coherent component. The second part, $I_{d}$, is related to the diffuse intensity, which can be interpreted as the incoherent field component propagation, i.e., it represents the specific intensity of the zero mean field. It is created by the scattering of the coherent component.
(8)$$\begin{array}{c} I\left(0;\theta ;\phi \right)=I_{\mathrm{ri}}\left(z;\theta ;\phi \right)+I_{d}\left(0;\theta ;\phi \right). \end{array}$$

We split the diffuse intensity into two parts:(9)$$\begin{array}{c} I_{d}\left(z;\theta ;\phi \right)=I_{1}\left(z;\theta ;\phi \right)+I_{2}\left(z;\theta \right), \end{array}$$
where $I_{1}$ is determined primarily by the forward lobe of the scatter function, and $I_{2}$ is the reminder of $I_{d}$ connected with the isotropic background.

The solution of the radiative transfer equation is the following:(10)$$\begin{array}{c} \frac{P_{R}\left(\tau ;\mu_{R};\phi_{R}\right)}{P_{\max}}=\underset{\mathrm{Coherent}\;\mathrm{component}\;\mathrm{attenuation}\;\mathrm{and}\;\mathrm{scattering}}{\underset{︸}{\exp\left(-{\left(\frac{\gamma_{\mathrm{RP}}}{\Delta \gamma_{R}}\right)}^{2}-\frac{\tau }{\mu_{p}}\right)}}\\ \underset{I_{1}\;\mathrm{incoherent}\;\mathrm{forward}\;\mathrm{scattering}}{\underset{︸}{+\frac{\Delta \gamma_{R}^{2}}{4}\{\left[\exp\left(-\frac{\hat{\tau }}{\mu_{P}}\right)-\exp\left(-\frac{\hat{\tau }}{\mu_{P}}\right)\right]{\bar{q}}_{M}\left(\gamma_{\mathrm{RP}}\right)}}\\ \underset{I_{1}\;\mathrm{incoherent}\;\mathrm{forward}\;\mathrm{scattering}}{\underset{︸}{+\exp\left(-\frac{\tau }{\mu_{p}}\right)\sum_{m=1}^{M}\frac{1}{m!}{\left(\frac{\alpha W_{\tau }}{\mu_{p}}\right)}^{m}\left[{\bar{q}}_{m}\left(\gamma_{PR}\right)-{\bar{q}}_{M}\left(\gamma_{PR}\right)\right]\}}}\\ \underset{I_{2}\;\mathrm{incoherent}\;\mathrm{isotropic}\;\mathrm{background}\;\mathrm{scattering}}{\underset{︸}{+\frac{\Delta \gamma_{R}^{2}}{2}\{-\exp\left(-\frac{\hat{\tau }}{\mu_{p}}\right)\frac{1}{P_{N}}}}\underset{I_{1}\;\mathrm{incoherent}\;\mathrm{isotropic}\;\mathrm{background}\;\mathrm{scattering}}{\underset{︸}{+\sum_{k=\frac{N+1}{2}}^{N}\left[A_{K}\exp\left(-\frac{\hat{\tau }}{S_{k}}\right)\sum_{n=0}^{N}\frac{1}{1-\frac{u_{n}}{S_{k}}}\right]\},}} \end{array}$$
where
(11)$$\begin{array}{c} {\bar{q}}_{m}\left(\gamma_{PR}\right)=\frac{4}{\Delta_{\gamma R}^{2}+m\Delta_{\gamma S}^{2}}\exp\left(-\frac{\gamma_{PR}^{2}}{\Delta_{\gamma R}^{2}+m\Delta_{\gamma S}^{2}}\right), \end{array}$$
(12)$$\begin{array}{c} P_{\mathrm{Max}}=P_{R}\left(0;\mu_{p},\phi_{p}\right)=\frac{\lambda_{o}^{2}}{4\pi }G_{R}\left(0\right)S_{p}. \end{array}$$

The attenuation coefficients $S_{k}$ are determined by the following characteristic equation:(13)$$\begin{array}{c} \frac{\hat{W}}{2}\cdot \sum_{n=0}^{N}\frac{P_{n}}{1-\frac{\mu_{n}}{S}}, \end{array}$$
where *N* is the odd integer chosen as a compromise for computing time, $P_{n}=\sin\left(\frac{\pi }{N}\right)\sin\left(\frac{n\pi }{N}\right)$, for (n = 1, …, N − 1) and $\hat{W}$ = $\frac{(1-\alpha )W}{1-\alpha W}$. Note that $S_{0,\ldots ,N/2}=-S_{N,\ldots ,(N+1)/2}$. The amplitude factors $A_{k}$ are determined by following system of linear equations:(14)$$\begin{array}{c} \sum_{k=\frac{N+1}{2}}^{N}\frac{A_{k}}{1-\frac{\mu_{n}}{S_{k}}}=\frac{\delta_{n}}{P_{n}}, \end{array}$$
where
(15)$$\begin{array}{c} \mu_{n}=-\cos\left(\frac{n\pi }{N}\right), \end{array}$$
(16)$$\begin{array}{c} \delta_{n}=\left\{\begin{array}{cc} \delta_{n}=0, & \mathrm{for}\;n\neq N\\ \delta_{n}=1, & \mathrm{for}\;n=N \end{array}\right. \end{array}$$

Here, we define all the parameters used in Equation (Equation 10):α (the ratio of the forward scattered power to the total scattered power),$\Delta \gamma_{R}$ (the beamwidth of the receiving antenna (degrees)),$\Delta \gamma_{R}$ = $0.6\cdot \Delta \gamma_{3\mathrm{dB}}$ (the introduced normalization for the beamwidth of the receiving antenna),$\gamma_{RP}$ (the angle between the receiving antenna point and *z* axis. One can assume 0 degree for perfect alignment with the *z* axis),$\Delta \gamma_{s}$ (3 dB width of the forward scatter lobe of the phase function),τ = ($\sigma_{a}$ + $\sigma_{s}$)z (it represents the medium’s optical density),W = $\frac{\sigma_{s}}{\sigma_{a}+\sigma_{s}}$ (albedo, which describes the likelihood of a scattering event in the homogeneous medium),$\hat{W}$ = $\frac{(1-\alpha )W}{1-\alpha W}$ (it specifies that only the fraction 1−α of the total power scattered in any scattering event is transferred into the isotropic background),$\hat{\tau }$ = τ(1 − αW) (it shows that for incoherent propagation, the contribution of the scattering term is reduced. The incoherent component is mainly affected by attenuation),μ=cos(θ) (It specifies the theta angle distance to normal *z*),$\mu_{p}=\cos\left(\theta_{p}\right)$ (it specifies the theta angle distance to normal *z* for incident intensity),$P_{N}=\sin^{2}\left(\frac{\pi }{2N}\right)$ (it is the normalization for piece-wise linear approximation for $I_{2}$),*m* order of the first term $I_{1}$ will not change significantly for m>10 (for most cases, M = 10),$G_{R}$ receiving antenna gain (for the specific angle).

### 2.2. Tree Alley Perspective

The radiative transfer equation can be considered with the assumption that the entire signal is transferred *through* the foliage with its absorption and emission as stated in Equation (Equation 10). However, as depicted above, the incoherent component may cause a change in direction, and the signal may go out from the trees with a randomized direction, unlike the coherent component. Then, as shown in Figure 3, diffracted or reflected components may emerge.

#### 2.2.1. Diffraction at Tree Edge

The wave that impinges on the edge of the tree corona can be diffracted. Then, it can travel along the tree corona and be diffracted at the second corona edge. It is possible that the signal transmitted by the antenna is diffracted on four sides of the tree: top and bottom, left and right. Furthermore, in the tree path, the signal can propagate along the edge of the entire alley.

In that case, the diffraction of the signal can also be modeled as a phenomenon occurring twice. Then, the well-known knife-edge diffraction model [22] given by Formula (Equation 17) can be extended to the two-edge diffraction model. Diffraction loss for a single edge is
(17)$$C_{\mathrm{diff}}=6.9+\log_{10}\left(\sqrt{{\left(v-0.1\right)}^{2}+1}+v-0.1\right),$$
where $C_{\mathrm{diff}}$ is the diffraction loss in dB, which is the approximation of a complex Fresnel integral valid for *v* greater than −0.78. *v* is the parameter depending on the wavelength λ and geometry of the obstacle with height *h* and diffraction angle ψ:(18)$$v=\sqrt{\frac{2h\psi }{\lambda }}.$$

In case diffraction is present at two edges, the double-edge diffraction loss ($C_{d\text{-}\mathrm{diff}}$) method can be applied from [23]. Then, the total loss of diffraction is
(19)$$C_{d\text{-}\mathrm{diff}}=C_{1}+C_{2}+C_{\mathrm{corr}},$$
where $C_{1}$ and $C_{2}$ are the diffraction losses for the first and the second edges, respectively. $C_{corr}$ is the correction coefficient since in the case of low separation between edges, the loss $C_{\mathrm{ddfif}}$ is underestimated [24]. Then the correction coefficient can be found as [17]
(20)$$C_{\mathrm{corr}}=10\log_{10}\left(\frac{\left(d_{1}+d_{2}\right)\left(d_{2}+d_{3}\right)}{d_{2}\left(d_{1}+d_{2}+d_{3}\right)}\right),$$
where $d_{1},d_{2}$, and $d_{3}$ are specific distances given in Figure 4. In addition, the parameter $h_{2}^{\prime }$ in Figure 4 is the height of the obstruction above the line of sight of the signal diffracted from the highest obstruction.

#### 2.2.2. Possible Ground Reflection

In addition, in some cases, ground reflection may play an important role, especially when small antenna height, long distance, and wide beamwidth of the antennas are observed. The reflection coefficient Γ then depends on the incident/reflection angle β and the relative permittivity of the medium η:(21)$$\Gamma =\frac{\cos\beta -\sqrt{\eta -\sin^{2}\beta }}{\cos\beta +\sqrt{\eta -\sin^{2}\beta }}.$$

The ground reflection loss for the reflected path with distance $d_{\mathrm{ref}}$ of the reflected path is then equal to
(22)$$L_{\mathrm{reflection}}=20\log_{10}\left(\frac{d_{\mathrm{ref}}}{\Gamma }\right).$$

### 2.3. Statistical Models

#### 2.3.1. Exponential Model with Maximum Attenuation (MA)

Foliage attenuation can be calculated with the exponential model with maximum attenuation, known as the MA model [11,17]. In that model, the distance $d_{in}$ is the distance in a homogeneous environment, as presented in Figure 5. The foliage is assumed to be homogeneous from the edge of the forest to the point of the receiving antenna. Then, the excess foliage loss $A_{F}$ is given as
(23)$$A_{F}=A_{m}\left(1+e^{-d_{in}\gamma_{v}/A_{m}}\right),$$
where $A_{m}$ and $\gamma_{v}$ are two crucial parameters which are foliage-specific and are determined by measurements. $A_{m}$ is the maximum attenuation per given depth in the foliage, and $\gamma_{v}$ is the species-specific attenuation coefficient per single meter of foliage given in [dB/m]. In the model, it is assumed that both Tx and Rx are of the same height; therefore, the signal path is parallel to the ground.

#### 2.3.2. Maximum Exponential Decay Model (MED)

Unlike maximum attenuation (MA), the maximum exponential decay model (MED) takes into account the frequency of the signal *f* given in MHz. The $d_{\mathrm{in}}$ measurements in the previous model represent the distance that the wave travels through the foliage, given in meters. The above two coefficients are weighted with a,b,c parameters, which are found with measurements:(24)$$L_{F}=a\cdot f^{b}\cdot d_{in}^{c}.$$

The example of parameter values found for the COST-235 model [25] is as follows: a=13.77,b=−0.009,c=0.26.

## 3. Measurement Campaign

The main motivation to carry out measurements of the foliage loss was to perform measurements between consecutive trees. The measurement scenario is the regular tree alley. We also performed a comparable measurement before the first tree. Then, we also compared our measurement results with statistical models and discussed the scattered-through (based on the RET model) and diffracted components.

### 3.1. Measurement Setup

Measurements were carried out in the vicinity of the Poznan University of Technology Lecture Center. In Figure 6, the satellite view of the measurement scenario is shown. Before the building on the square, the trees are planted in a regular grid, with a precise distance of 6m between consecutive trees in the row. Although six rows of trees can be found in the courtyard, two rows were selected for measurements and marked as Scenarios 1 and 2. The trees in all rows are homogeneous in terms of species. Furthermore, the distance between rows and consecutive trees is regular. The distance between rows is 3m, while the distance between trees in a row is 6m.

The crucial novelty of the measurements was to check tree attenuation when placing both antennas in the same horizontal plane for the alley of trees. Moreover, such a regular order of planted trees allowed for verifying attenuation introduced by at least one and up to six trees in the row. Then, the considered measurement setup was to place the Tx antenna at the end of the tree row at a height of 2.2m, then the Rx at the same height between the tree coronas. The directional horn antennas SL-WDPHN-1840-1719-K were used. The half-power beam width of the antennas was 16 degrees at the carrier frequency of 28GHz, while the antenna gain for that frequency was 18dBi.

The measurement equipment connection was as follows (Figure 7): Continuous wave (CW) was generated by an Anritsu MG3690C at frequency $f_{c}=\left\{26,28,30,32\right\}\;\mathrm{GHz}$. The transmitted power was 0dBm. The signal was received by an Anritsu MS2720T spectrum analyzer. The noise floor during the measurement was −130dBm, and the dynamic range was at least 106dB in 1Hz resolution bandwidth (RBW).

Measurements at all points were automated with Matlab script and remote mode for the measurement devices. Such a configuration allowed us to (i) take measurements for many frequencies and (ii) take multiple measured values per each point, then use the median value as the main one.

The illustration of the measurement scenario is shown in Figure 8. The Tx position for all measurements was constant, 4m away from the first tree in the row. Then, the Rx position during the measurements was placed in different free-space points between trees.

### 3.2. Single-Tree Model

The initial measurements carried out at the measurement site show that the foliage attenuation results are related to the tree size in the alley. This conclusion led to the creation of the single-tree model. The generic model of a park tree is shown in Figure 9. In that model, the height of the corona of the tree is $C_{h}$, and the diameter of the corona is $C_{w}$. The total height of the tree is $T_{h}$, and the height of the trunk is $T_{h}-C_{h}$. The measurement site is shown in Figure 10. The precise sizes of the trees can be found in Table 1 and Table 2. Not only dimensions of trees but also the predicted volume of tree corona can be found there. Most trees have a corona volume from 7 to $16\;m^{3}$. There are two exceptions: the third tree for Scenario 1 with a volume of nearly $50\;m^{3}$ and the fifth tree in Scenario 2. Tree coronas have greater variability; the diameter of the corona varies from 1.81m to 3.8m, while the height of the tree ranges from 4.5m to 6.38m.

## 4. Results of Measurements

Measurements were conducted for Scenarios 1 and 2; for each Scenario, the other set of trees was taken into account. Then, in each scenario, the same set of trees was considered, but two directions of measurement were considered. Precisely, the transmit antenna was set in two variants at two ends of the tree row highlighted in the legend as ‘measurement 1’ and ‘measurement 2’. In Figure 11 and Figure 12, the results of the measurements are presented for the two scenarios for four carrier frequencies. Foliage attenuation is there presented in the function of foliage depth. Therefore, the point ‘zero’ at the depth of the foliage is the beginning of the first tree in the tree alley. In each measurement scenario, Tx was located 4 m away from the first tree corona, as presented in Figure 8.

The general observation of the measured foliage attenuation is that the greatest increase in attenuation is between the second and third trees in the alley. This is in line with the exponential character of the loss described precisely in Section 2.

In Figure 13, Figure 14, Figure 15, Figure 16, Figure 17, Figure 18, Figure 19 and, Figure 20, the measurements are presented and divided by frequency and scenario. They are compared with three reference models (COST-235, Weissberger, and FITU-R) and two statistical models with tuned parameters, according to frequency. Maximum attenuation is presented as ‘Model 1’ and has $A_{m}$ and γ tuned, while MED, named in the figures as ‘Model 2’, has three parameters tuned. In addition, in the legend of each plot, the value of the root mean square error is given.

In Figure 21 and Figure 22, all measurements from Scenario 1 with parameter fitting can be found. The MA and MED models were found according to the minimum root mean square error (RMSE) and the minimum median of squared errors (LMedS). The minimum median error in Scenario 2 was found for the MA model (0.73dB) and in Scenario 2 for the MED model (1.75dB).

## 5. Discussion

### 5.1. Measurement Campaign Results Analysis

Combining the theory presented in Section 2 with the results of the measurement campaign, one can observe a few facts.

The geometry of the tree, including its density and distance between it and the next trees, has a significant impact on signal attenuation. The scattering of the coherent component is the dominant contributor (Table 3) when dealing with this type of vegetation area. A possible explanation for these large $\left(\sigma_{a}+\sigma_{s}\right)$ values can be derived from the fact that in a foliage environment, all scattering obstacles are large compared to the millimeter wavelength. Behind large obstructions such as dense corona, deep shadows exist, and since the campaign was carried out over a path through maximum vegetation density, it is very likely that the RX antenna was placed in the shadow zones. Therefore, the measured data may indicate a higher value for $\left(\sigma_{a}+\sigma_{s}\right)$ than one would obtain on average (measured in all spatial directions and by averaging the power received on these paths).

It was observed that younger London plane trees could be a more serious obstacle when it comes to radio propagation attenuation. Additionally, it is important to mention that for a young London plane tree, the most characteristic element is that the value of the density of leaves is very high. This means that the tree canopy is clumped, creating highly scattered layers.

It should be understood that the comparison of theory and experiments involves a certain amount of conjecture since the measured curves, as opposed to the smooth theoretical ones, show significant fluctuations. The RTE model has a priori assumptions and simplifications and is not fully identical to the experimental setup. It should be emphasized that the theoretical model applies to a forest half-space illuminated by a plane wave, whilst the incident beam has a finite cross section and trees have finite height. This implies that losses due to radiation into the ground and air regions are not accounted for in the RTE model.

In order to visualize these relationships, the comparison of the RTE theoretical values and measurements is presented in Figure 23 and Figure 24.

Further study is required in that field to seek novel methods that can simplify the mathematical description of phenomena or maintain the same level of complexity but may allow one to describe the huge number of different scenarios based on the theoretical model approach. This would be useful from the perspective of high-band RF planning for cellular networks.

### 5.2. Wider Application of the Proposed Vegetation Loss Model

Let us emphasize that the main focus of this work was to concentrate on fixed wireless access links in the vicinity of urban buildings. In such a case, the straight-line propagation through trees is one of the possible situations where the findings discussed in this paper can be applied directly. However, it is worth considering the wider application context of the presented measurements. In particular, we suggest that the measured vegetation loss could be considered as an additional component in the overall calculation of the total link loss (as part of a link budget calculation). For example, the vegetation loss model proposed in this manuscript could be directly included in simplified models such as the Free Space PL model or similar, which reflect mainly the change of the path loss as a function of distance or frequency only; to some extent, the vegetation loss can be included as the additive components in more detailed models as well; however, more careful analysis would be beneficial to not overestimate the impact of vegetation loss on signal propagation. In the context of the FWA link, vegetation loss can be applied straightforwardly; while designing the point-to-point link, appropriate budget calculations have to be performed. When there is no natural obstacle (such as a tree), well-known models can be applied that do not consider the presence of vegetation loss. Similar conclusions can be drawn for the winter period, where the presence of trees is not that critical in path-loss modeling as there is no foliage. However, when one is aware of the presence of trees (and in particular knows the type of tree, their number, etc.), the measured vegetation loss can be included in the calculated link budget as an additive factor (in logarithmic scale).

### 5.3. Impact of the Corona Shape

Finally, it is worth noting that the proposed model has been tuned based on the measurements, where triangle-like corona shapes were present. This means that this model is well adjusted to the specific tree shape, i.e., trees with a triangle-like shape of the corona. One may foresee that the measurement with other corona shapes (or with trees with different shapes at all) may result in different tuning parameters. In consequence, the model discussed in this manuscript can generalize only to some extent, i.e., one has to remember that the tuning coefficient may be different when other corona shapes are considered. The application of the proposed model could be treated as some sort of coarse approximation of the foliage attenuation for the FWA scheme. In principle, fine-tuned models could be proposed for different corona types. However, this is a space for interesting new investigations in that domain. Moreover, while referring directly to the electromagnetic scattering theorem, in our work, we made the assumption that the Gaussian phase function is considered for random scatterers. In fact, more research is required to specify the exact phase function for specific corona tree shapes. It is another emerging area of research where we foresee that physics-based machine learning could be used. However, one also has to remember that the other contribution made in this paper is the RTE analysis, where the results do not depend on the shape at all.

## 6. Conclusions

The experimental data shown in this article indicate a high attenuation rate at short distances into woods and the sharp transition that exists between the lower and higher attenuation regions. This transition occurs after the second tree in the present measurement campaign. Each tree’s corona is characterized by quite a large value of density, understood as the number of scatterers per unit volume. Certainly, measured attenuation exceeds the values of the single-tree attenuation model. This is why this phenomenon requires deeper investigation. The enhanced model would replace the simplified forest half-space by a two-layer slab model on a ground plane with well-defined absorption and scattering parameters.

The major observation during the present campaign is that the measured woodland is not fully statistically homogeneous. There are irregularities in the canopy of trees and free space between trees. The LIDAR method seems promising when it comes to an accurate description of a single tree’s physical structure. Using this method, one can quantify the wood area and acquire proper tree dimensions and the volume occupied by leaves.

In this article, the model for scattering through foliage is described. However, the model is quite complicated, especially when the correct input parameters are needed. Some research and development steps were carried out to improve this approach. Furthermore, as a future step, to simplify this task, the idea of obtaining branches and leaves’ parameters may be developed using a drone equipped with a LIDAR scanner. Drone flight around the obstacle on the radio-wave path can quantify a sufficient number of cloud points. Based on this, important tree parameters can be extracted and recalculated. This method is capable of introducing a modern digital combined solution to the mmWave measurement methodology.

## Figures and Tables

**Figure 1 sensors-24-03190-f001:**
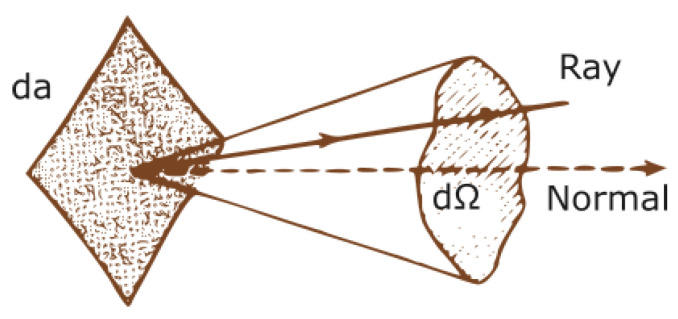
Geometry for normally incident rays.

**Figure 2 sensors-24-03190-f002:**
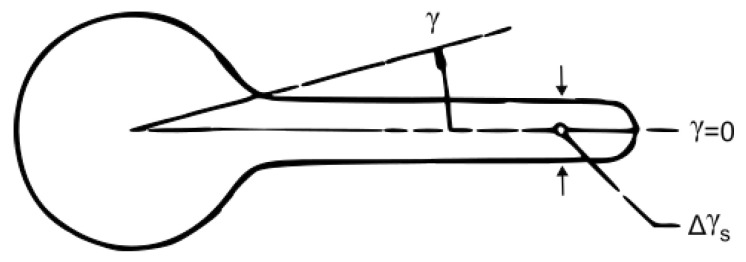
Phase function definition (isotropic background and narrow forward lobe) where p(γ) represents (angle) deviation to the main axis and $\Delta \gamma_{s}$ is the beamwidth of the Gaussian lobe.

**Figure 3 sensors-24-03190-f003:**
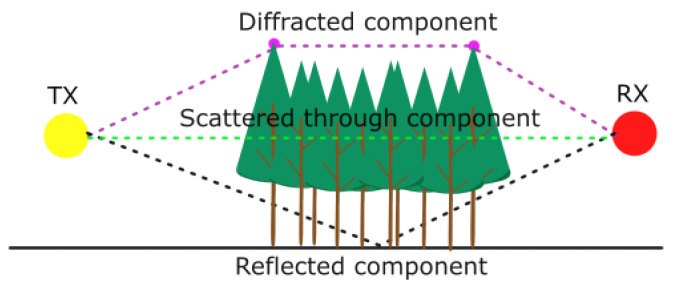
Considered modeling for foliage attenuation.

**Figure 4 sensors-24-03190-f004:**
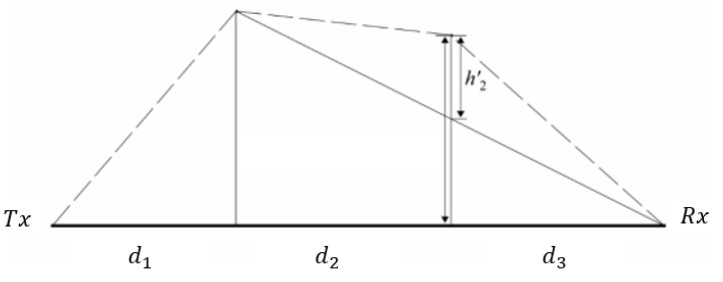
Diffraction model for double edges.

**Figure 5 sensors-24-03190-f005:**
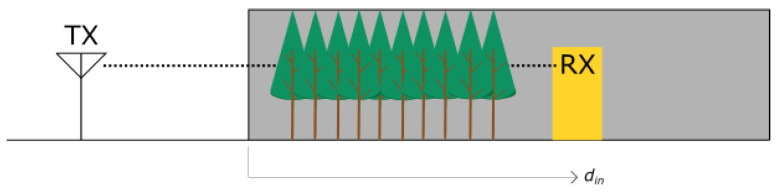
Homogeneous woodland of length $d_{\mathrm{in}}$ at the link between transmitter (TX) and receiver (RX). The woodland in the link is represented by a gray color, while yellow represents the position of the Rx.

**Figure 6 sensors-24-03190-f006:**
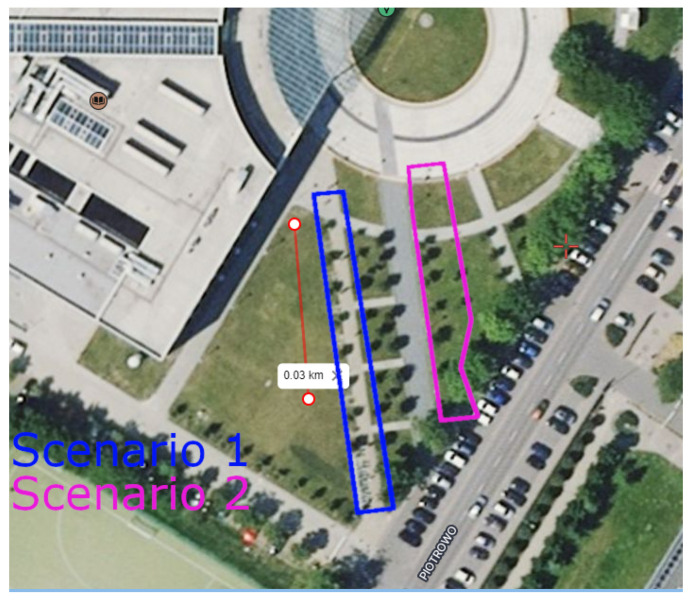
Satellite view of the area with considered measurement scenarios. Rectangular areas with trees for two scenarios are marked in two colors. Furthermore, next to Scenario 1, there is a line of length 30m as the scale.

**Figure 7 sensors-24-03190-f007:**
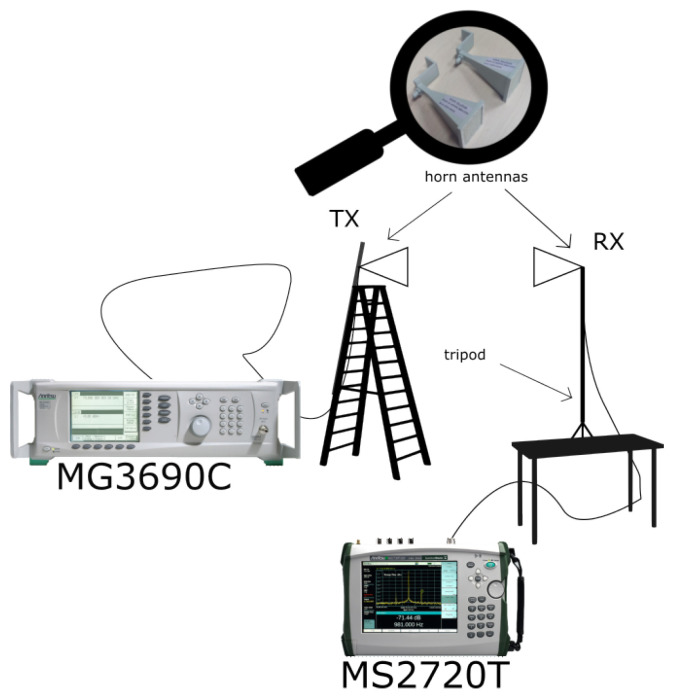
Measurement equipment configuration overview.

**Figure 8 sensors-24-03190-f008:**
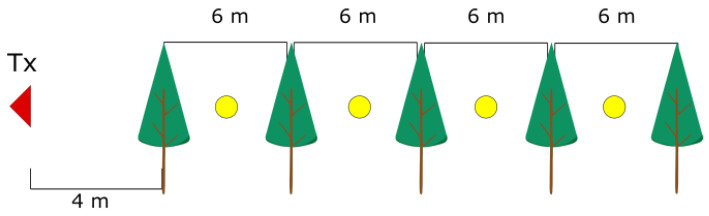
Measurement scenario with tree alley. The Tx (red triangle) is located 4m from first tree at the front, while the distance between trees in the alley is 6m. The Rx (yellow circle) is placed between tree coronas; the distance between the trees in the alley is 6m.

**Figure 9 sensors-24-03190-f009:**
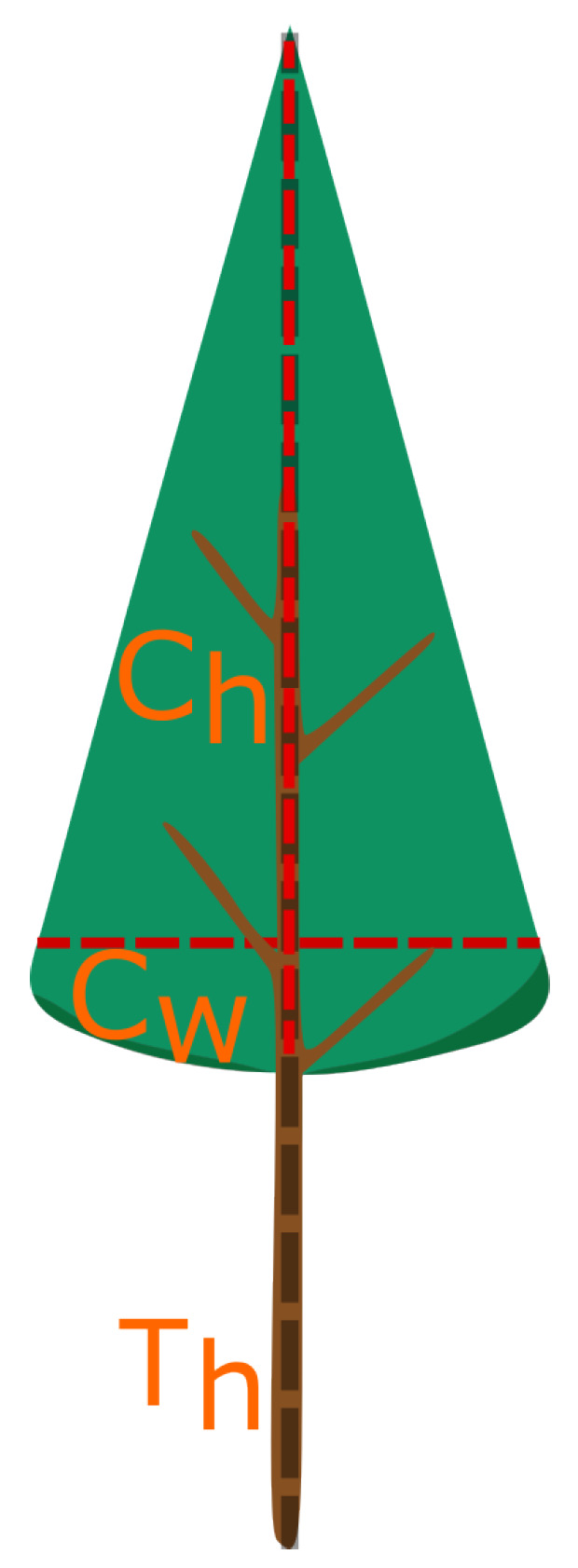
Single-tree model. Each tree has its height $T_{h}$, corona height $C_{h}$, and corona width $C_{w}$.

**Figure 10 sensors-24-03190-f010:**
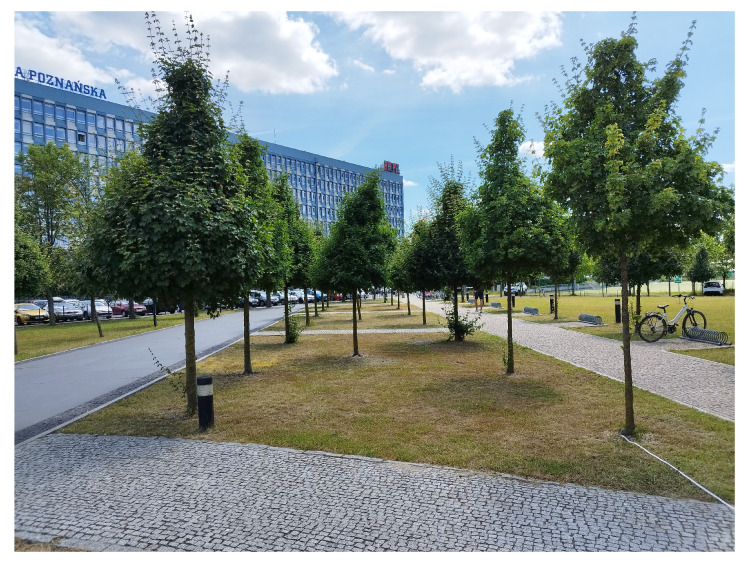
Measurement site at Poznan University of Technology.

**Figure 11 sensors-24-03190-f011:**
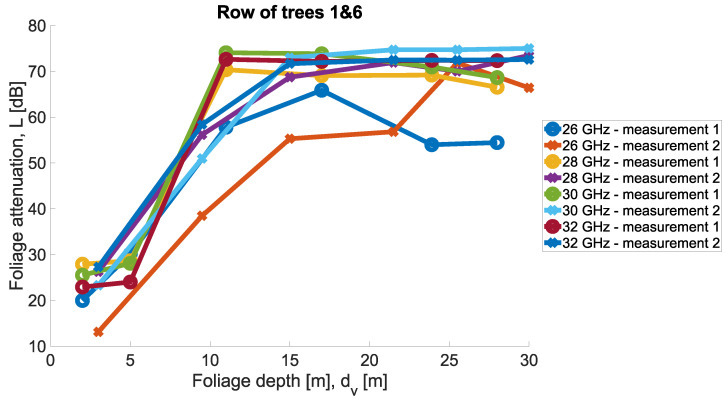
Foliage attenuation measurements for Scenario 1.

**Figure 12 sensors-24-03190-f012:**
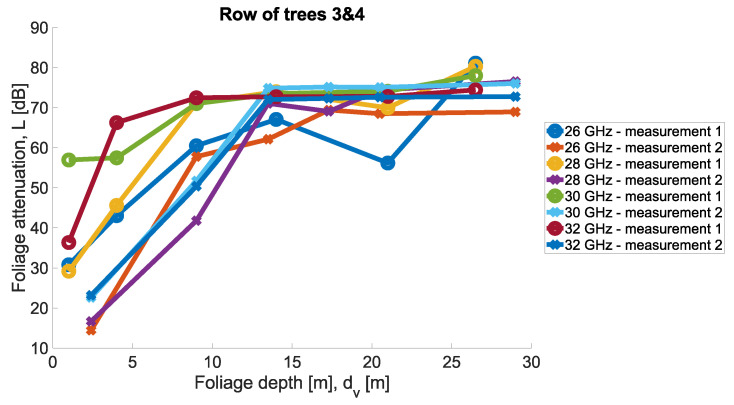
Foliage attenuation for Scenario 2.

**Figure 13 sensors-24-03190-f013:**
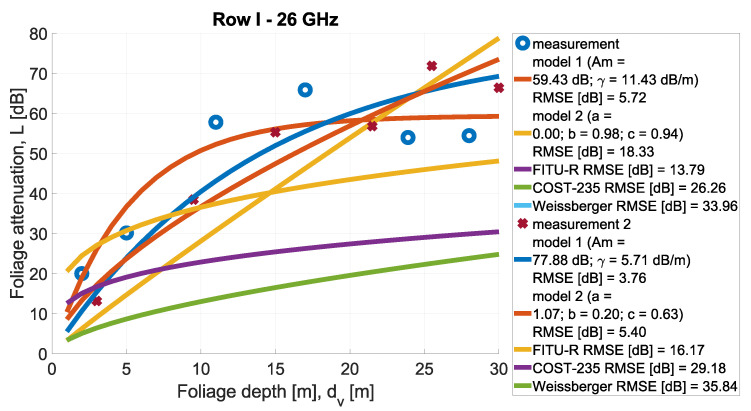
Comparison of measurements and statistical models in Scenario 1 at frequency 26 GHz.

**Figure 14 sensors-24-03190-f014:**
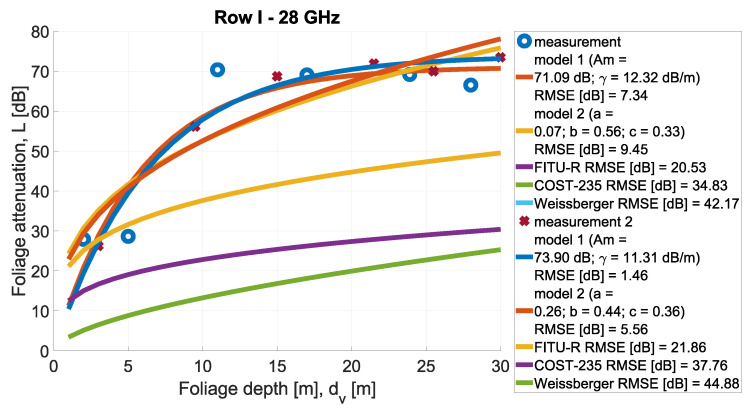
Comparison of measurements and statistical models in Scenario 1 at frequency 28 GHz.

**Figure 15 sensors-24-03190-f015:**
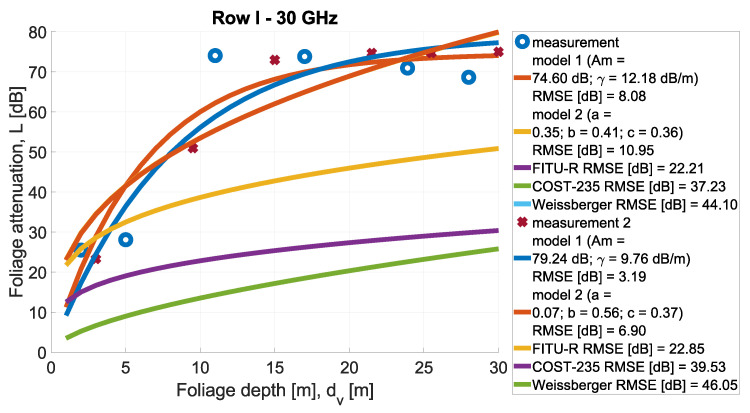
Comparison of measurements and statistical models in Scenario 1 at frequency 30 GHz.

**Figure 16 sensors-24-03190-f016:**
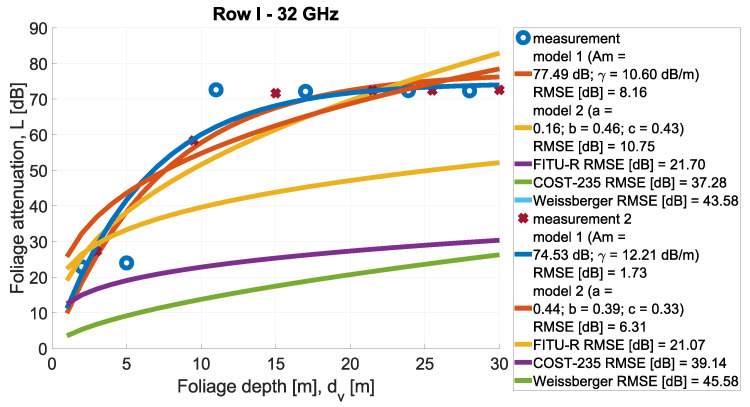
Comparison of measurements and statistical models in Scenario 1 at frequency 32 GHz.

**Figure 17 sensors-24-03190-f017:**
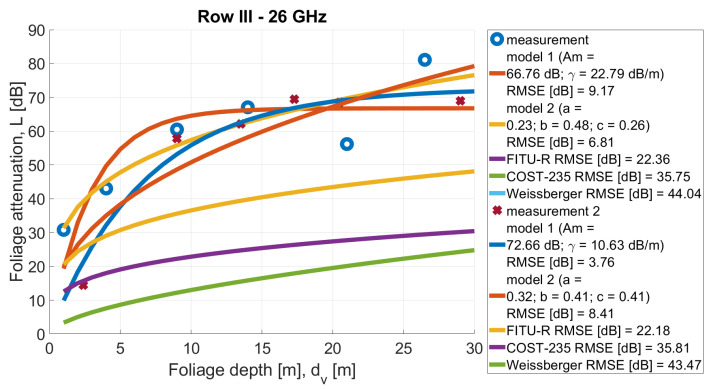
Comparison of measurements and statistical models in Scenario 2 at frequency 26 GHz.

**Figure 18 sensors-24-03190-f018:**
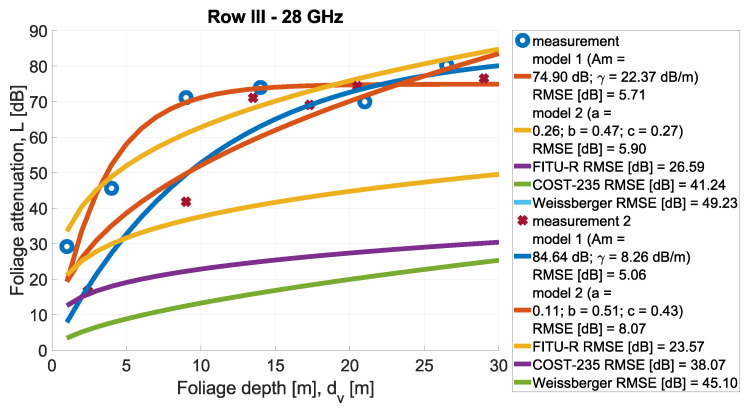
Comparison of measurements and statistical models in Scenario 2 at frequency 28 GHz.

**Figure 19 sensors-24-03190-f019:**
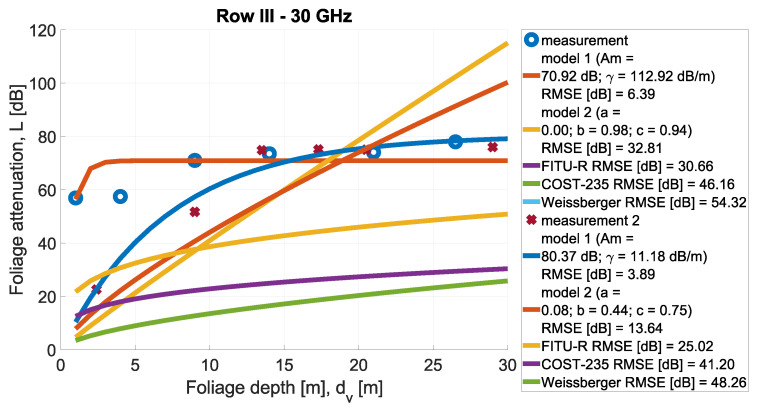
Comparison of measurements and statistical models in Scenario 2 at frequency 30 GHz.

**Figure 20 sensors-24-03190-f020:**
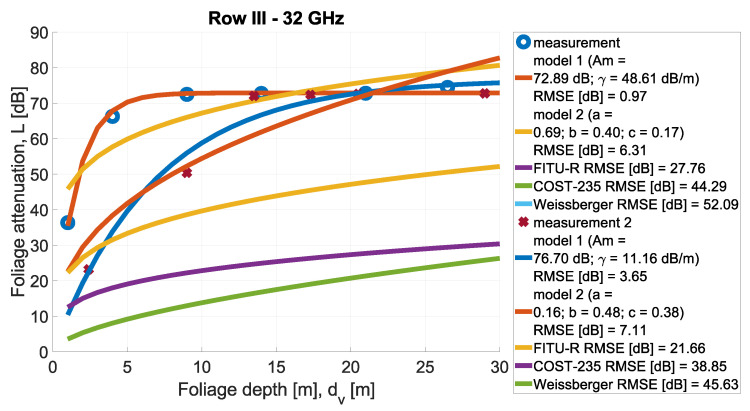
Comparison of measurements and statistical models in Scenario 2 at frequency 32 GHz.

**Figure 21 sensors-24-03190-f021:**
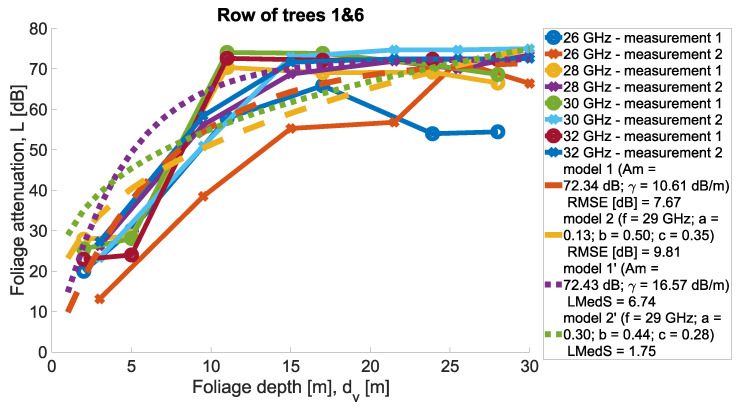
All measurements and tuned models for Scenario 1.

**Figure 22 sensors-24-03190-f022:**
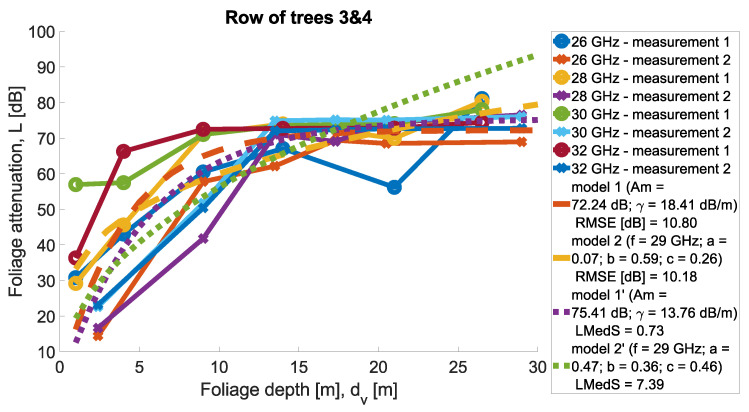
All measurements and tuned models for Scenario 2.

**Figure 23 sensors-24-03190-f023:**
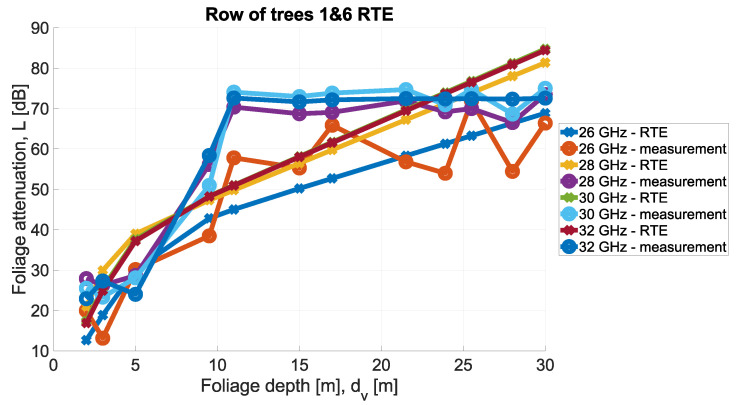
RTE vs. measurements—comparison.

**Figure 24 sensors-24-03190-f024:**
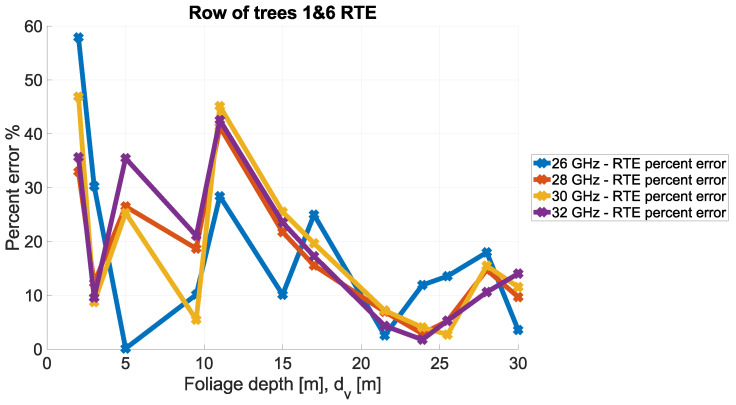
RTE vs. measurements—difference (percent error).

**Table 1 sensors-24-03190-t001:** Precise corona and tree sizes in Scenario 1.

Tree ID	$T_{H}\;\left[m\right]$	$C_{W}\;\left[m\right]$	$C_{H}\;\left[m\right]$	$C_{\mathrm{vol}\_\mathrm{pred}}\;\left[m^{3}\right]$
1	4.6	2.1	2.57	8.90
2	4.89	2.4	3	13.57
3	6.38	3.8	4.4	49.90
4	5	2.4	3.5	15.83
5	4.6	2.46	3.2	15.20
6	4.56	2	3	9.42

**Table 2 sensors-24-03190-t002:** Precise corona and tree sizes in Scenario 2.

Tree ID	$T_{H}\;\left[m\right]$	$C_{W}\;\left[m\right]$	$C_{H}\;\left[m\right]$	$C_{\mathrm{vol}\_\mathrm{pred}}\;\left[m^{3}\right]$
1	4.93	2.53	3.15	15.83
2	4.73	2.23	3	11.71
3	5.26	2.35	3.22	13.97
4	4.5	1.81	2.82	7.26
5	5.3	3	3.75	26.5
6	4.6	2.13	3	10.69

**Table 3 sensors-24-03190-t003:** Tuning performed on the RTE solution using the data gathered during Scenario 1.

RTE Fitted Parameters	α	$\gamma_{s}$	*W*	$\left(\sigma_{a}+\sigma_{s}\right)$
26 GHz measurement	0.8366	359.9737	0.9634	1.4743
28 GHz measurement	0.8827	357.6472	0.9585	2.4831
30 GHz measurement	0.9509	356.9840	0.8401	2.0325
32 GHz measurement	0.8462	359.7209	0.9392	1.9761

## Data Availability

Data are contained within the article.

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
