# Peer review of "Vegetation Loss Measurements for Single Alley Trees in Millimeter-Wave Bands"

_sensors, 2024, doi:10.3390/s24103190_

Round 1

Reviewer 1 Report

Comments and Suggestions for Authors

1. Why does the model assumes the TX and RX devices are at the same level as the trees? Usually the base station antennas are placed higher and facing down, can the authors justify the motivation here?

2. If the goal is to model the vegetation loss in straight line transmission of signals, how can this model be utilized to predict more common urban environment (for example, with buildings).

3. It would be good to add a simulation model and show how the signals attenuate when passing through the trees.

4. The results only shows attenuation in dB, but can you add results to show real-life impact to the communication links with this kind of attenuation, or explain in more details?

5. The plots from figure 11 onward need to be improved for better clarity. For example, wider lines and symbols. It's hard to compared the results right now. 

Reviewer 2 Report

Comments and Suggestions for Authors

The paper reports a study on vegetation attenuation for a single tree alley for 2632 GHz, the radiative transfer equation and experimental measurements are all presented. The authors give detailed theoretical analysis and experimental testing procedures and results. This work is benefit for millimeter wave wireless access research, and I believe this paper will be interesting for the readers.

However, I consider there is a critical problem in the analysis, thus, I suggest author making some necessary revisions for the manuscript. The critical problem is that the tree model used in analysis is corona, but the shape of the trees in Fig. 10 is triangle. Based on the electromagnetic scattering theory, different shape would cause different scattering. Then, the tuning parameters proposed for models maybe not caused by the frequency increasing, but by the tree shape different. Author should give more discussions or clarification for this difference.

Additionally, the words in Fig. 13-22 are not clear, I suggest enlarges the words size.

Reviewer 3 Report

Comments and Suggestions for Authors

I find your research of interest. However, there are some things to improve:

Why are the frequencies 26, 28, 30, and 32 Ghz selected?

It is necessary to calculate and plot the difference between RET and the measurements.

Why not use FDTD for modeling?

highlight the advantages of your model over others

And the attenuation due to vegetation was to be expected.

Reviewer 4 Report

Comments and Suggestions for Authors

In the manuscript “Vegetation Loss Measurements for Single Alley Trees at Millimeter Waves Bandwidth”, Cichon and co-workers discuss a vegetation attenuation measurement at millimeter-wave frequencies (26-32 GHz). The authors provide a comprehensive theoretical background on modeling vegetation attenuation using the Radiative Transfer Equation (RTE) and various statistical models. The measurement campaign was conducted in a tree alley scenario, with the transmitter and receiver placed at the same height between tree crowns. The results show that vegetation loss increases significantly after the second tree in the alley, which aligns with the exponential nature of the attenuation described by the RTE model.

Overall, the paper is well-written and provides valuable insights into vegetation attenuation modelling and measurements, but there is room for further exploration and clarification on some aspects of the work. Please see below the following comments:

1.        The authors mention the potential impact of diffraction at tree edges and ground reflections, but they do not provide a quantitative analysis of these effects. It would be interesting to see how these factors contribute to the overall attenuation and how they could be incorporated into the models.

2.        The paper focuses on a specific tree species (London Plane trees). It would be valuable to investigate how the results and proposed tuning parameters might vary for different tree species or foliage densities (at least numerically).

3.        The measurements were conducted at a fixed antenna height of 2.2 meters. It would be interesting to explore the impact of varying the antenna heights on the measured attenuation, as this could be relevant for different FWA deployment scenarios.

4.        The paper could benefit from more discussion on the practical implications of the findings for FWA network planning and deployment in urban or suburban environments with tree alleys or similar vegetation scenarios. I suggest the authors to add something in the introduction or conclusion section.

5.        On similar note I suggest the authors to add some relevant recent work in the literature on millimetre and terahertz sensing under complex propagation; see Refs [1, 5].

References 

[1]              L. Leibov et al., “Speckle patterns formed by broadband terahertz radiation and their applications for ghost imaging,” Sci Rep, vol. 11, no. 1, Art. no. 1, Oct. 2021, doi: 10.1038/s41598-021-99508-1.

[2]              H. Shen et al., “Spinning disk for compressive imaging,” Opt. Lett., OL, vol. 37, no. 1, pp. 46–48, Jan. 2012, doi: 10.1364/OL.37.000046.

[3]              V. Cecconi, V. Kumar, J. Bertolotti, L. Peters, A. Cutrona, L. Olivieri, A. Pasquazi, J.S. Totero Gongora, and M. Peccianti ACS Photonics 2024 11 (2), 362-368 DOI: 10.1021/acsphotonics.3c01671

[4]              J. Zhao, Y. E, K. Williams, X.-C. Zhang, and R. W. Boyd, “Spatial sampling of terahertz fields with sub-wavelength accuracy via probe-beam encoding,” Light Sci Appl, vol. 8, no. 1, p. 55, Dec. 2019, doi: 10.1038/s41377-019-0166-6.

[5]              R. I. Stantchev et al., “Noninvasive, near-field terahertz imaging of hidden objects using a single-pixel detector,” Sci. Adv., vol. 2, no. 6, p. e1600190, Jun. 2016, doi: 10.1126/sciadv.1600190. 

Comments on the Quality of English Language

minor typos

Round 2

Reviewer 4 Report

Comments and Suggestions for Authors

I thank the authors for having replied to all my questions and clarifying some doubts about the manuscript. 

Author Response

Thank you for the positive feadback.